# OpenReview forum: "INVESTIGATING ANNOTATOR BIAS IN LARGE LANGUAGE MODELS FOR HATE SPEECH DETECTION"
_NeurIPS.cc/2024/Workshop/SafeGenAi — SafeGenAi Poster_

### Official Review · Reviewer_huRt · 2024-10-09
**Please validate and refine the HateBiasNet dataset**

**Rating:** 5
**Confidence:** 5

**Review:**

The paper addresses an important and relevant topic and provides valuable insights into annotator biases in LLMs for hate speech detection. The research methodology is well-structured and the findings contribute to the existing literature. However, there are some limitations and areas for improvement that should be addressed.

1. Could you provide more details on the evaluation of the HateBiasNet dataset? What evaluation metrics were used, and how does it compare to existing hate speech detection datasets? How do you plan to further validate and refine the HateBiasNet dataset? Are there any plans to release an updated version of the dataset with more annotations?
2. Can you provide more details about the annotation process, including the guidelines provided to the annotators and the measures taken to ensure consistency and clarity in the annotation process? Have you considered the impact of other factors, such as cultural background or geographical location, on annotator biases in LLMs? How do these factors interact with the biases related to gender, race, religion, and disability?
3. How did you select the specific biases and vulnerable groups to analyze in this study? Were there any specific criteria or considerations that guided your selection?
4. Can you provide more insights into the potential consequences of these biases in real-world applications, such as hate speech detection systems or content moderation platforms? How can these biases perpetuate harmful stereotypes and discrimination?

---

### Official Review · Reviewer_m6pq · 2024-10-09
**The HateBiasNet dataset underscores the importance of addressing biases in LLMs for equitable AI, highlighting methodological weaknesses and the need for more comprehensive related works.**

**Rating:** 6
**Confidence:** 5

**Review:**

Investigating Annotator Bias in Large Language Models for Hate Speech Detection

This paper explores the biases inherent in Large Language Models (LLMs) when annotating hate speech data. The authors identify four key categories of bias—gender, race, religion, and disability—across four LLMs: GPT-3.5, GPT-4o, Llama-3.1, and Gemma2. They create a custom dataset, HateBiasNet, to study these biases, conducting a comparative analysis with the ETHOS dataset. The paper seeks to elucidate the implications of these biases for model performance and societal impact, especially concerning vulnerable groups. The authors argue that understanding these biases is crucial for developing equitable AI systems in hate speech detection. The paper makes a significant contribution by highlighting the biases present in LLMs during hate speech detection, emphasizing the importance of understanding these biases to foster responsible AI practices. It adds to the growing body of literature focused on bias in AI systems and provides a foundation for future research aimed at addressing these challenges.

Strengths

The paper demonstrates significant strengths by addressing a critical issue in AI ethics and natural language processing, specifically focusing on biases in AI systems that can perpetuate discrimination against marginalized communities, making it timely and relevant. The research questions are clearly articulated, concentrating on the existence, contributing factors, and potential mitigations for annotator bias in LLMs, which enhances the study's focus. Additionally, the authors adopt a comprehensive approach, analyzing multiple dimensions of bias and employing rigorous methodologies, including a systematic analysis of biases across different LLMs, complemented by a comparative analysis of datasets. Furthermore, the development of the HateBiasNet dataset is a valuable contribution to future research in hate speech detection, filling a gap in existing datasets and facilitating a deeper examination of bias in this domain.

Weaknesses

1.	Lack of Methodological Transparency: While the paper describes the methodology, it could benefit from more detailed explanations of the data collection and annotation processes. For example, how the annotators were selected, their qualifications, and the exact criteria used for categorizing tweets into 'hateful' or 'not hateful' could enhance reproducibility and credibility.
2.	Overemphasis on LLMs: The paper primarily focuses on LLMs as annotators, potentially downplaying the role of traditional human annotators and their biases in the annotation process. A more balanced view could include a discussion on how LLMs compare to human annotators in terms of performance and bias.
3.	Generalizability of Findings: The findings may not be fully generalizable to contexts outside hate speech detection. The specific nature of hate speech and the demographic characteristics of the annotators may limit the applicability of the results to other domains of NLP.
4.	Insufficient Mitigation Strategies: While the authors discuss potential solutions to mitigate bias, the recommendations lack specificity. More concrete strategies or frameworks could be proposed, emphasizing how researchers can practically implement these mitigations in their work.
5.	Narrow Related Works: The related works section lacks citations of relevant publications, particularly those similar to Basile et al. (2019), Zampieri et al. (2019, 2020), and Waseem (2016). Key works, such as Ravi & Vela (2024), Ravi et al. (2023), Mahata et al. (2019), and Behzadan et al. (2018), should be included as they provide important insights into implicit offensive language, including ideological and extreme bias, toxic comments, targeted offenses, violent threats, and bias indicators relevant to this paper.
o	Ravi, K., & Vela, A. E. (2024). Comprehensive dataset of user-submitted articles with ideological and extreme bias from Reddit. Data in Brief, 56, 110849.
o	Ravi, K., Vela, A. E., Jenaway, E., & Windisch, S. (2023, December). Exploring Multi-Level Threats in Telegram Data with AI-Human Annotation: A Preliminary Study. In 2023 International Conference on Machine Learning and Applications (ICMLA) (pp. 1520-1527). IEEE.
o	Vaidya, A., Mai, F., & Ning, Y. (2020, May). Empirical analysis of multi-task learning for reducing identity bias in toxic comment detection. In Proceedings of the International AAAI Conference on Web and Social Media (Vol. 14, pp. 683-693).
o	Mahata, D., Zhang, H., Uppal, K., Kumar, Y., Shah, R., Shahid, S., ... & Anand, S. (2019, June). MIDAS at SemEval-2019 task 6: Identifying offensive posts and targeted offense from Twitter. In Proceedings of the 13th International Workshop on Semantic Evaluation (pp. 683-690).
o	Hammer, H. L., Riegler, M. A., Øvrelid, L., & Velldal, E. (2019, September). Threat: A large annotated corpus for detection of violent threats. In 2019 International Conference on Content-Based Multimedia Indexing (CBMI) (pp. 1-5). IEEE.
o	Behzadan, V., Aguirre, C., Bose, A., & Hsu, W. (2018, December). Corpus and deep learning classifier for the collection of cyber threat indicators in Twitter stream. In 2018 IEEE International Conference on Big Data (Big Data) (pp. 5002-5007). IEEE.
6.	Complementary Insights: This work closely explores themes found in another submission, "OFFENSIVELANG: A Community-Based Implicit Offensive Language Dataset," in terms of task and methodology. Both "OFFENSIVELANG" and "Investigating Annotator Bias in Large Language Models for Hate Speech Detection" examine themes of offensive language, bias in LLMs, and dataset creation, contributing to discussions on managing harmful content. However, "OFFENSIVELANG" focuses on offensive content generation by LLMs and implicit bias, while "Investigating Annotator Bias" examines the biases LLMs exhibit in annotating hate speech. Despite some overlap, the papers differ in their methodologies and scope, making them complementary rather than redundant. Clear differentiation of their unique contributions can help avoid concerns about overlap.

Conclusion

Overall, the paper "Investigating Annotator Bias in Large Language Models for Hate Speech Detection" makes a substantial contribution to understanding the biases present in LLMs, particularly in the context of hate speech annotation. By identifying key categories of bias and creating the HateBiasNet dataset, the authors provide valuable insights that enhance the discourse on responsible AI practices. Despite some methodological weaknesses, such as the lack of transparency in data collection processes and insufficient mitigation strategies, the research addresses a critical area of AI ethics. Moreover, the related works section lacks citations of relevant publications, particularly those similar to the findings of Basile et al. (2019) and Zampieri et al. (2019, 2020), as well as key works by Ravi et al. (2024), Ravi et al. (2023), Mahata et al. (2019), and Behzadan et al. (2018). Addressing these gaps could further strengthen the foundation for future research aimed at understanding bias in LLMs. Additionally, the complementary relationship between this paper and others like "OFFENSIVELANG" underscores the need for clear differentiation of their unique contributions to avoid concerns about overlap.

---

### Official Review · Reviewer_UY1P · 2024-10-09
**Review of INVESTIGATING ANNOTATOR BIAS IN LARGE LANGUAGE MODELS FOR HATE SPEECH DETECTION**

**Rating:** 4
**Confidence:** 5

**Review:**

Pros:

[1] The authors present a new hate speech dataset that can be used to evaluate the annotation biases in the LLMs

[2] The authors compare the annotations with human annotations for their dataset, hence grounding the dataset

Cons:

[1] The methodology used to compare the annotations by LLMs with human annotations is slightly flawed. By assigning a "persona" to the LLM, the authors provide additional context, which changes the LLM's decisions. By not including or mentioning the same demographics for their human annotator counterparts, any claims of comparing the two are lost. To ensure that their results are grounded, the authors must compare a "target" LLM annotator to its "target" human annotator, to ensure reliability in their results. T

[2] Table [3] presents the accuracy of the model when compared to human annotations. While this is not an incorrect metric, it is incorrectly claimed as "bias". To identify a bias, the authors must compare this to the ground truth of the model itself, i.e. when the model does not provide any context or persona. Any deviation by the addition of a persona, hence would be claimed as a bias.

[3] The authors do not explain their reasoning for using "Not Asian" and "Not Black" as individual personas. There are no characteristics associated to being "Not Asian" or "Not Black". This indicates the model to be weak in terms of characterizing the features and, hence results are not controllable and may not follow a set pattern across the dataset. It would be advisable for the models to be either clearer in their persona description or present an explanation/reference for their choice. It would also be interesting to see the comparison within races, for example, "Asian" vs "Black" and see how the model is biased in this scenario (as done in works such as: Gupta, Shashank, et al. "Bias runs deep: Implicit reasoning biases in persona-assigned llms." arXiv preprint arXiv:2311.04892 (2023).)